# Blue-Shifting Mechanofluorochromic Luminescent Behavior of Polymer Composite Films Using Gelable Mechanoresponsive Compound

**Mizuho Kondo \*, Yuya Morita and Nobuhiro Kawatsuki \***

Department of Applied Chemistry, Graduate School of Engineering, University of Hyogo, 2167 Shosha Himeji, Hyogo 671-2280, Japan; et21c002@steng.u-hyogo.ac.jp
\* Correspondence: mizuho-k@eng.u-hyogo.ac.jp (M.K.); kawatuki@eng.u-hyogo.ac.jp (N.K.);
Tel.: +81-79-267-4014 (N.K.)

**Abstract:** Mechanochromic luminescent dyes change their luminescence color upon exposure to external mechanical stimuli. In this study, we synthesized a liquid crystalline mechanochromic luminescent dye containing a terminal cholesterol molecule. The dissolution of the dye in 1,4-dioxane resulted in the formation of a gel. The luminescence of the xerogel obtained from the dioxane solution changed from green to blue upon grinding, indicating mechanochromic luminescence behavior. The anisotropic patterning of short-wavelength-shifted luminescence color change by directional handwriting on surface layer of liquid crystal was successfully demonstrated. Furthermore, blue-shifting mechanoresponsive polymer composite surface was fabricated by using the luminophore.

**Keywords:** mechanofluorochromic dye; liquid crystal; gel





## 1. Introduction

Mechanochromic luminescence (MCL) is a phenomenon in which the photoluminescence (PL) color of a material changes in response to the mechanical stimuli (such as compression, stretching, shearing, bending, impact, and friction). MCL compounds utilize the macroscopic mechanical stimuli and result in microscopic optical properties in a stimulated area without affecting the size or shape of the MCL object. Therefore, MCL compounds can be introduced into various materials and can be applied in optical recordings [1], security inks [2], memory devices [3,4], and novel diagnostic tools [5,6]. It is a relatively new research field that has been actively studied over the last decade, and a variety of materials have been investigated, including crystals, liquid crystals (LCs) [7], gels [8,9], and polymers [10,11]. In the case of LCs, anisotropy and aggregation structures can be easily controlled by external stimuli. By combining these properties with the functions of MCL, it is possible to develop special MCL materials that exhibit multicolor emission [12–14], circularly polarized PL [15], and anisotropy. Previously, we studied MCL materials based on liquid crystalline cyanostilbenes and detected the grinding direction using the stress orientation of LCs [16,17] and emission color control using supramolecular LC complexes [18,19]. Recently, rod-shaped tolane-terminated cyanostilbene was prepared, and its mechanoresponsive change in luminescent behavior was evaluated. The luminescence color of the dye depended on the precipitation method, and the PL spectrum shifted to either longer or shorter wavelengths upon grinding. When the solution was recrystallized by removing the solvent, the color shifted from blue to green. In contrast, when the solution was precipitated by dropping it into a poor solvent, the color shifted from yellow to green indicating that multiple polishing responses can be induced from a single material [20]. Since dyes with cyanostilbene as a luminescent backbone exhibit various intra- and intermolecular interactions [21], it can be presumed that any or some of them may act during the formation of solids.

Since solids exhibiting short wavelength shifts can be obtained by precipitation, a technique that preferentially produces solids with low crystallinity, we focused on low-molecular-weight gels to stably produce thin films. Low-molecular-weight gels are materials with a continuous structure of macroscopic dimensions composed of a low concentration of gelling agent and solvent and exhibit solid-like rheological behavior. Gels formed by low-molecular-weight gelators can switch between flowing sols and solid-state gels depending on the external stimuli. In addition, gels retain their internal structure even after the solvent is dried; thus, thin films that inherit the solid structure on precipitation are formed by depositing them as gels and forming films. Such low-molecular-weight gels are developed by introducing long hydrophobic tails, such as cholesterol, alkyl, and alkoxy chains. In particular, cholesterol can form stable gels by the synergistic effect of cooperative non-covalent bonding. Thus, vigorous research has been conducted, and application development aimed at optical function and collection performance has been explored in materials with cholesterol molecule [22]. Conversely, cholesterol has a rigid ring-fused structure and alkyl chain; therefore, it can be used as a substituent to introduce LC properties, and gelation ability can be expected without impairing the LC properties. Tolane-terminated cyanostilbene produces stable LCs [23], and the liquid crystallinity of cholesterol can be effectively utilized, but no research has been conducted with regard to liquid crystallinity. In this study, we prepared gelable liquid crystalline cyanostilbene with terminal cholesterol molecule (**2ECh)** and investigated the change in the luminescence color of the prepared mechanochromic dye upon gelation.

## 2. Materials and Methods

The synthesis of **2ECh** is shown in Scheme 1. All starting materials and solvents were used as-received from Tokyo Kasei Chemicals and Aldrich Co. The polymeric composite was prepared using Poly (methyl methacrylate) purchased from Aldrich Co (182230-500G, Mw: 120,000, Tg: 90 °C) without any purification.

**Scheme 1.** Schematic synthesis of **2ECh**.

### 2.1. Synthesis of **1**

Compound **1** was synthesized according to a previously reported procedure [20].

[1]H NMR (400 MHz, DMSO): δ (ppm) 8.18 (s, 1H), 8.05–7.99 (m, 5H), 7.81 (d, $J$ = 5.9 Hz, 2H), 7.65 (d, $J$ = 8.2 Hz, 2H), 7.47 (d, $J$ = 2.3 Hz, 2H), 7.25 (d, $J$ = 5.9 Hz, 2H), 2.60 (q, $J$ = 6.4 Hz, 4H), and 1.15 (t, $J$ = 7.5 Hz, 3H).

IR (KBr, cm$^{-1}$): 2965, 2217, 1683, 1607, 1511, 1419, 1286, 1196, and 1125.

### 2.2. Synthesis of 2ECh

Compound **1** (1.1 g, 3 mmol), cholesterol (1.2 g, 3 mmol), and a trace amount of 4-dimethylaminopyridine were dissolved in tetrahydrofuran (30 mL). N, N'-dicyclohexylcarbodiimide (0.6 g, 3 mmol) was added dropwise to the solution while stirring, and the reaction mixture was stirred at room temperature (20–25 °C) for 16 h. The resulting suspension was evaporated, and the residue was dissolved in dichloromethane. The suspension was filtered to remove urea crystals. After the solvent was evaporated, the residue was purified by column chromatography (silica gel, eluent: chloroform). The resulting solid was recrystallized from chloroform/hexane to obtain **2ECh** as a yellowish-green solid (0.23 g, M.p. 217 °C, 6.3% yield).

[1]H NMR (400 MHz, CDCl$_3$): δ (ppm) 8.14-8.12 (m, 2H), 7.93 (d, J = 8.7 Hz, 2H), 7.69–7.67 (m, 2H), 7.59 (m, 3H), 7.47-7.45 (m, 2H), 7.19 (d, J = 8.7 Hz, 2H), 5.42 (d, J = 3.2 Hz, 1H), 4.88 (s, 1H), 2.67 (q, J = 7.6 Hz, 2H), 2.47 (d, J = 7.8 Hz, 2H), 2.04-0.85 (m, 46H), 0.68 (s, 3H).

$^{13}$C NMR (101 MHz, CDCl$_3$): δ (ppm) 145.3, 140.9, 139.6, 137.5, 133.4, 132.4, 132.3, 131.8, 130.2, 129.2, 128.1, 126.1, 125.1, 123.0, 120.0, 113.4, 88.1, 75.1, 56.8, 56.2, 50.1, 42.4, 39.8, 39.6, 38.3, 37.1, 36.7, 36.3, 35.9, 32.0, 32.0, 29.0, 28.3, 28.1, 27.9, 24.4, 23.9, 22.9, 22.7, 21.1, 19.5, 18.8, 15.4, 12.0

IR (KBr, cm$^{-1}$): 3412, 2936, 2220, 1715, 1467, 1496, 1415, 1370, 1277, 1187, and 1112.

*2.3. Equipment*

The synthesized compounds were characterized using Fourier-transform nuclear magnetic resonance (FT-NMR; JEOL JNM-ECZ 400 MHz) (Tokyo, Japan) and infrared (FT-IR; JASCO FT-IR 6000) (Tokyo, Japan) spectroscopies. $^1$H and $^{13}$C NMR spectra were recorded for CDCl$_3$ and DMSO-d$_6$ at room temperature using tetramethylsilane (TMS; δ 0.00) as an internal standard. The mesomorphic properties were evaluated using a polarizing optical microscope (Olympus, BH50) (Tokyo, Japan) and differential scanning calorimetry (DSC, Hitachi High-Tech Science DSC7020) (Tokyo, Japan). PL spectra were collected using a luminescence spectrometer (Hitachi, F-4500) (Tokyo, Japan). The solid-state quantum yield of the luminophore was measured using a fluorescence spectrometer equipped with an integrating sphere (JASCO FP-6600) (Tokyo, Japan) with an excitation at 360 nm. The PL lifetimes of the composite films were measured using a nanosecond spectrofluorometer (Horiba, FluoroCube) (Kyoto, Japan) and an excitation wavelength of 370 nm. Powder X-ray diffraction (XRD) measurements were performed using a diffractometer (Rigaku SmartLab 3 kW) (Tokyo, Japan) with a standard parallel beam setup. The microscopic images of the xerogel were observed using a scanning electron microscope (Keyence V8800) (Osaka, Japan). The surface profile of the films was evaluated with an interferometric surface profiler (Ryoka Systems, VertScan R3300H) (Osaka, Japan). Thin layers for spectroscopy and XRD were prepared from dioxane solution on glass substrates using spin casting method. Thin layer of luminophore or polymer composite was ground by pen and the luminophore powder was ground using agate mortar.

## 3. Results and Discussion

The chemical structure of **2ECh** is shown in Figure 1. Cyanostilbene, a typical MCL substituent, has a rod-like molecular shape with a high affinity for LC materials. In addition, the π-conjugated structure was extended by introducing tolane, which is widely used as a photoemissive mesogen substituent. The same structure has been used for the mechanoresponsive parts and tends to form antiparallel molecular pairs in crystals. The presence of cholesterol introduces intermolecular interactions; thus, one-dimensional bonds are expected to contribute to gel formation [24]. The DSC thermograph and polarized optical micrograph obtained at 260 °C are shown in Figure 2a. The polarized light micrograph shows typical oily stripes, indicating that the compound exhibits a chiral nematic phase from 217 °C to >300 °C. The LC properties were maintained at temperatures above 300 °C, and no clearing point was observed before decomposition. However, the compound in the LC phase showed a yellow melt and no obvious selective reflection was observed (Figure 2b).

The gelation ability of the dye was examined at room temperature. The mixture of **2ECh** and solvent was heated up to the boiling point of the solvent (101 °C) to obtain a homogeneous solution. Then, the solution was cooled to room temperature, and the gelation ability of the dye was examined using an inversion test. The results are summarized in Table 1. Cholesterol-incorporated compounds can gel with a wide range of materials, but **2ECh** gelled only at a high concentration of 10 mg/mL **2ECh** in dioxane. It has been reported that intermolecular interactions between the alkyl groups (van der Waals forces) and hydrogen bonds are important for the formation of molecular assemblies [25]. **2ECh** has a short alkyl chain, and thus, it is assumed that its gelation ability is not high.

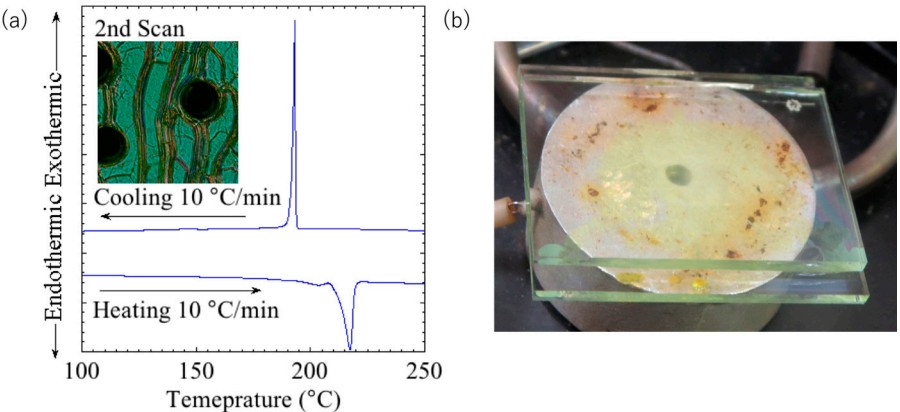

**Figure 1.** Chemical structure of the compound used in this study.

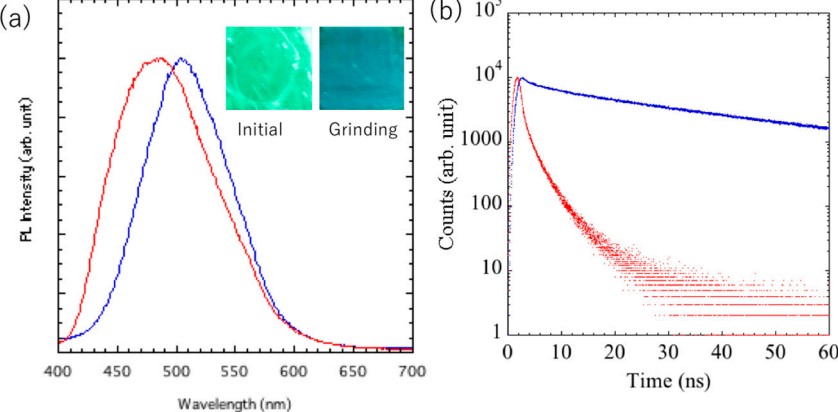

**Figure 2.** (**a**) Differential scanning calorimetry thermogram of **2ECh** (inset: polarized optical microscopy image of **2ECh** at 250 °C). (**b**) photograph of **2ECh** at 250 °C injected in 8-µm cell.

**Table 1.** Gel-formation tests for 2ECh in various solvents.

|  | 1,4-Dioxane | Ethyl Acetate | Hexane | Methanol | Tetrahydrofuran |
|---|---|---|---|---|---|
| 5 mg/mL | PG | I | I | I | S |
| 10 mg/mL | G | I | I | I | S |

PG: Partially Gelled; G: Gelled; I: Insoluble; S: Solution at 20 °C.

The gels formed by dioxane were air-dried, and the mechanoresponsive behavior was examined in the thin-film state. The PL spectrum and fluorescence lifetime changes are shown in Figure 3a and b, respectively. It has been reported that materials whose color changed by gelation retained their luminescent color even in xerogels after the solvent was removed and showed a grinding response [26]. The powder in its initial state exhibited blue luminescence and no response to grinding. When the dioxane solution was heated to a sol state and a thin film was formed by spin coating, a uniform green-emitting thin layer was obtained.

**Figure 3.** (**a**) Photoluminescence (PL) spectrum (**b**) and fluorescent decay profile of **2ECh** before (blue) and after (red) mechanical grinding. The inset in (**a**) displays PL color of **2ECh**.

In the initial state, the thin layer of the luminophore showed a maximum emission wavelength, fluorescence lifetime, and quantum yield of 500 nm, 35.0 ns, and 4%, re-

spectively, and after mechanical grinding, the values shifted to 470 nm, 1.3 ns, and 2%, respectively. The fluorescence lifetime was longer (~10 ns) than that of the conventional tolane-terminated cyanostilbene, suggesting the formation of an excimer. Figure 4a shows the XRD patterns of the initial and ground glass substrate coated with the luminophore. The initial film had no diffraction peaks and was amorphous before and after grinding. The xerogel showed the same mechanical response when obtained as a powder and ground using agate mortar. However, no obvious change such as size and structure were observed by mechanical grinding (Figure 4b). Accordingly, the microstructures of the xerogels were observed by scanning electron microscopy (SEM, Figure 4c). SEM demonstrated that the xerogel had micrometer-scale fibrous aggregates, whereas the grinding powder showed no such structure. From the changes in the SEM images, it can be presumed that the amorphous fibrous structure on the solid surface derived from the xerogel induces the green luminescence, and that the luminescence color changes when these are destroyed by mechanical grinding.

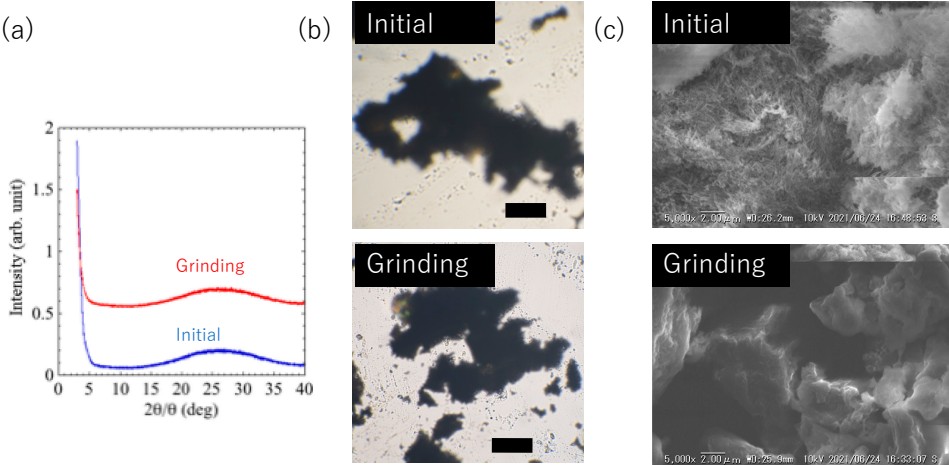

**Figure 4.** (**a**) X-ray diffraction (XRD) spectrum (**b**) optical image (**c**) and scanning electron microscopy (SEM) images of **2ECh** before and after mechanical grinding. The black line in (**b**) is scalebar for 0.8 mm.

The directional mechanochromic behavior of the **2ECh** film was studied. As shown in Figure 5a, "T" was written on the film with a pen, and when MCL was induced, blue luminescence in the shape of the letter was observed. This letter was formed by applying mechanical pressure in two orthogonal directions. Figure 5(b) shows a photographic image of the film obtained using the linear polarizer. When the polarization direction of the polarizer is parallel to the grinding direction, the grinding area shows a high PL intensity. Furthermore, the maximum wavelength of the PL spectrum was larger in the parallel direction than in the perpendicular direction, indicating that the wavelength was blue-shifted. Figure 5c shows the polarized PL spectrum of the polished film. The PL parallel to the grinding direction was larger than that in the perpendicular direction. The polarization ratio P was calculated using the equation, $P_{||}/P_{\perp}$, is up to 2.5, where $P_{||}$ and $P_{\perp}$ are the PL intensity at the respective maximum PL wavelengths measured with light polarized parallel and perpendicular to the grinding direction. The optical anisotropy was also observed by polarizing optical microscopy. As shown in Figure 5d, distinct bright lines are seen at 45° with respect to the analyzer by rotating the film, which confirms the polarized emission due to grinding. Figure 5e displays Surface profile of the luminophore layer. The xerogels were found to be aggregated and distributed in a speckled pattern to form chunks and approximate height of them are over 0.1 μm. These chunks were broken down and leveled out at ground area (upper green part of the figure). Since anisotropy is observed in the dye stretched into a paste at the ground area, it can be presumed that the stress orientation is partially inherited from the characteristics of liquid crystal.

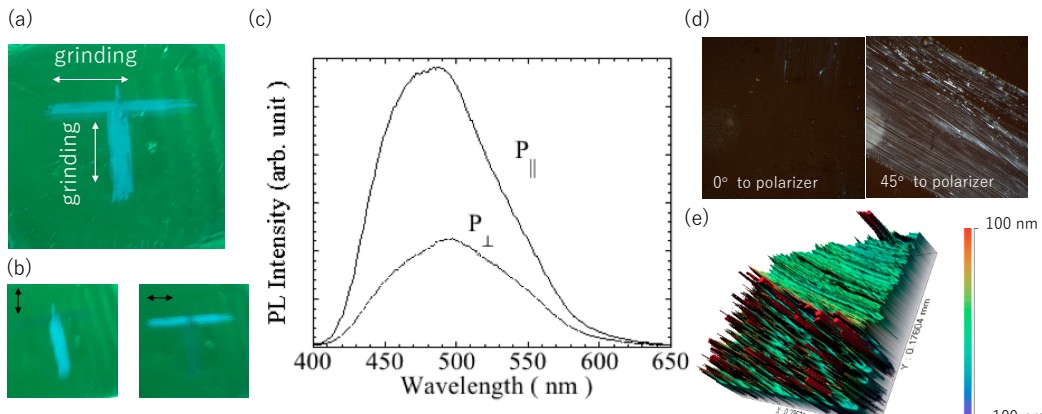

**Figure 5.** (**a**) Photograph of the patterned photoemission of a **2ECh** film following directional grinding. (**b**) Photographs obtained using a linear polarizer, (**c**) polarized PL spectra, (**d**) polarizing optical micrograph and (**e**) surface profile of the ground thin layer. The black arrows in (**b**) indicate the polarization direction. The subscripts in (**c**) designate the directions parallel (∣∣) and perpendicular (⊥) to the grinding direction.

The blue-shifted photoluminescence was maintained even when the glass substrate coated with the grinding luminophore was heated to 220 °C or lower. On the other hand, when heated above that temperature, green luminescence was again observed, and the luminescence color change could be repeated by additional grinding. As shown in Figure 6a, the maximum fluorescence wavelengths of the glass substrate coated with annealed and ground luminophore were 500 and 490 nm, respectively. Figure 6b shows the diffraction patterns of the annealed and post-grinding luminophore on glass substrate. The ground and heated thin films exhibited numerous diffraction peaks, indicating that they were crystalline. XRD results showed strong peaks in the 2θ = 3–30° region at 3.91, 4.71, 5.26, 8.50, 10.70, 13.43, 17.24, 18.61, 19.86, and 25.24°. These reflections were found to correspond to surface spacings of 22.6, 18.6, 16.8, 10.4, 8.3, 6.6, 5.1, 4.8, 4.5, and 3.5 Å, respectively. Among these spacings, the reflection at 3.5 Å is attributed to the distance between two π-stacked stilbene units in the self-assembled state. Further, the reflections at 4.8 and 4.5 Å are attributed to the presence of hydrogen-bonded β-sheet-like arrays of gelling agent molecules in the assembled gel state [24]. Grinding of this thin film resulted in a significant decrease in the diffraction intensity, but not a complete disappearance. However, the fluorescence lifetimes of the annealed and ground luminophore were 10 and 3 ns, respectively (Figure 6c). These results reveal that the color change was induced by heating, but the luminescent species were altered, and the color change in the thin film of the dye was irreversible for the dye alone.

Finally, to overcome the reproducibility drawback, a polymer composite film consisting of a gelling agent and a non-photoluminescent polymer was prepared. We have previously reported that thin films made of polymers induce area-selective grinding responses. Composites with polymers allow for easy preparation of thin films and acquisition of repeatable properties [16,17] however, the materials that could be thinned were limited to those that shifted to the long-wavelength side by grinding, as well as those that shifted to the short-wavelength side could not be fabricated. A blue-shifting MCL polymer composite film using an immiscible polymer matrix has been reported by Pucci et al. [27], but a grinding-responsive polymer composite using a miscible polymer has not been reported. A dioxane solution of a 1:1 (*w/w*) mixture of PMMA and dye was prepared and spincasted. Initially, the composite layer showed green luminescence with a maximum emission wavelength of 500 nm, and after grinding, it showed blue luminescence with a maximum emission wavelength of 470 nm. The ground film returned to its original wavelength when heated above 180 °C (Figure 7a). Figure 7b shows change in XRD profile after grinding and annealing. No new peaks were observed in the XRD patterns after grinding, suggesting that the film remained in the amorphous state. Upon heat treatment,

a slight increase in the peak at 4.71 and 5.26° was observed, but no clear crystal peak was observed. It was presumed that the dye dissolved in the polymer became sol-like, and then gelled again. Figure 7c shows surface profile of the composite layer with partial grinding. It was found that the layer was composed of the similar structure to the luminophore. The height of dispersed chunks is over 0.1μm and no PMMA surface was found on the surface, indicating the composite did not form a film. It can be presumed that the gelation of the luminophore with the polymer inhibited the film formation. The composite succeeded in inhibiting crystallization, but failed to form a film, so further investigation of the material is needed. Gelation in the coexistence of a gel and polymer is expected to result in a film with excellent mechanical stability, processability, and formability, and is capable of withstanding bulk loads, such as stretching and compression.

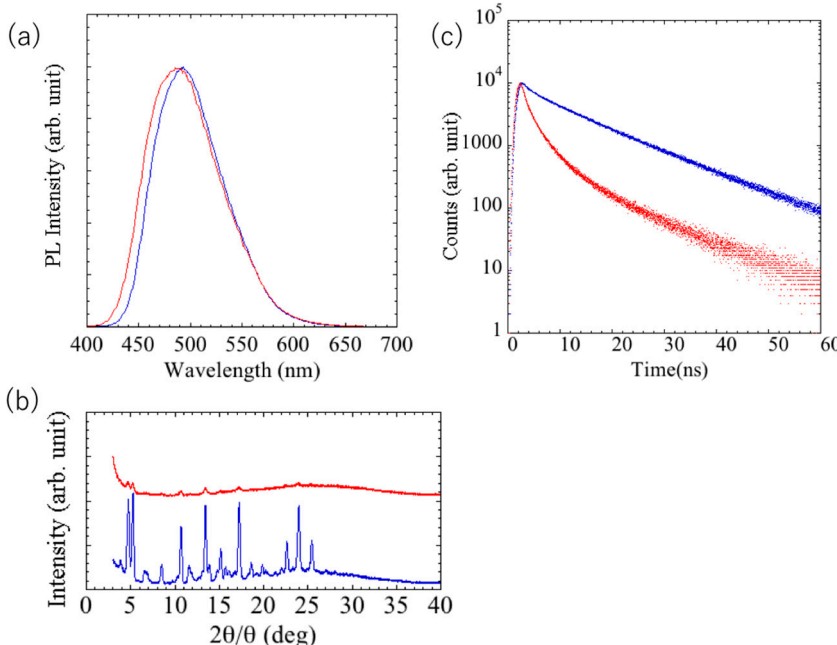

**Figure 6.** (**a**) PL spectrum, (**b**) XRD, and (**c**) fluorescent decay profile of recrystallized **2ECh** before (blue) and after (red) mechanical grinding.

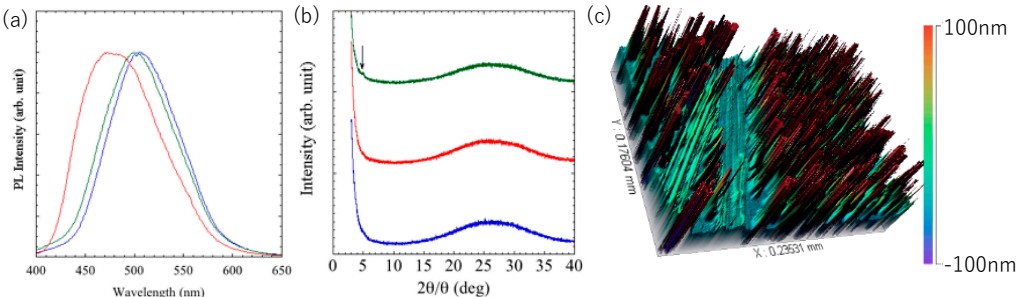

**Figure 7.** (**a**) Change in PL spectrum (**b**) and XRD pattern of **2ECh** dispersed in PMMA (1:1 (*w/w*)). Color of the spectrum indicates the film before grinding (blue), after grinding (red) and annealing (green) respectively. (**c**) Surface profile of the composite layer. The black arrow in (**b**) indicates the diffraction of **2ECh** crystal (see Figure 5b).

## 4. Conclusions

We synthesized a liquid crystalline mechano-fluorochromic dye containing a terminal cholesterol molecule and explored its mechanoresponsive behavior. Film formation was improved by gelation, and the processability and responsiveness were also improved when thin film was mixed with polymer. Cholesterol, a typical biocompatible functional group,

can be used to mimic lipid bilayers and interact with various functional materials in living organisms. In addition, cholesterol has many optically active sites; thus, it is expected to have extended functions beyond its use as a gelator, such as circularly polarized light emission and higher order LCs [28,29]. However, these functions are still under study.

**Author Contributions:** Conceptualization, N.K. and M.K.; methodology, M.K.; validation, Y.M., M.K.; investigation, Y.M.; resources, N.K.; writing—original draft preparation, M.K.; writing—review and editing, N.K.; visualization, M.K.; supervision, N.K. All authors have read and agreed to the published version of the manuscript.

**Funding:** This research received no external funding.

**Data Availability Statement:** Not applicable.

**Acknowledgments:** We are sincerely grateful to Kenji Iimura and Hiroshi Satone (University of Hyogo) for the SEM measurements.

**Conflicts of Interest:** The authors declare no conflict of interest.

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
