# Peer review of "Blue-Shifting Mechanofluorochromic Luminescent Behavior of Polymer Composite Films Using Gelable Mechanoresponsive Compound"

_crystals, doi:10.3390/cryst11080950_

Round 1
Reviewer 1 Report
See the attached file.

Author Response
- Despite the term "Liquid crystalline" is a part of the manuscript title, the liquid crystal phase description is very scanty. The temperature range of the liquid crystal phase is not indicated in the manuscript. A reader can only guess from thermogram (Fig.2) that LC phase is above 220 oC. The properties of the liquid crystal phase also absent. If above 220 oC there is a chiral LC phase then what is the natural pitch of chiral helix and chirality sign? Does the selective reflection exist in the visible spectral range? What is the optical anisotropy of the material in LC phase (difference of the principal refractive indices)? What is a role of the liquid crystalline phase in the described phenomenon of the "directional blue-shifting mechanofluorochromic" luminescence?. What is the temperature range for the "directional blue-shifting mechanofluorochromic" luminescence, and how the liquid crystalline properties influence on mechanofluorochromic luminescence, especially on account the LC phase exists only at rather high temperatures?
We have added the result that the liquid crystalline phase is observed from 220°C to the manuscript. In the section describing the change with heating, we added that the blue emission is maintained up to 220 °C.
The pitch direction is assumed to be right-handed based on the similarity with cholesterol-derived liquid crystals identified in the past (Y. Chen, P. Lu, Z. Li, Y. Yuan, Q. Ye, H. Zhang, ACS Appl. Mater. Interfaces, 2020, 12, 56604-56614), and the birefringence is expected to be higher than in the corresponding paper because of the introduction of a tolane, but this is not clear at present. Furthermore, since the temperature of the liquid crystalline phase is high and the liquid crystalline properties are hardly reflected at room temperature, we omitted the evaluation of these properties in this paper. However, the pitch control of the cholesteric phase is an interesting field because it affects selective reflection and circularly polarized light emission, and we plan to conduct further research in this area. Accordingly, we added two additional reference at conclusion that exploit circular polarized photoemission. As pointed out by the reviewer, the word "liquid crystal" was deleted from the title of the paper because the paper does not put forward the properties of liquid crystals.
The role of the liquid crystalline phase is not considered to have a direct effect, but it is not irrelevant. In our previous work, we have confirmed that anisotropy can be observed when liquid crystalline mechanoresponsive compounds are ground in the temperature range below the liquid crystalline phase (crystalline or glassy state). Since anisotropy is observed in the dye stretched into a paste, it can be presumed that partial aggregation of the liquid crystal affects the fracture direction of the solid, resulting in anisotropic collapse and stress orientation. However, since no diffraction information or other clues were available in this paper, we could only speculate and add the following phrase to our discussion.
"It can be presumed that the stress orientation is partially inherited from the characteristics of liquid crystal."
We also added the photograph of luminophore in 8-mm glass cell at 260 °C under natural light at Figure 2. The cell colored yellow and exhibited no obvious selective reflection at liquid crystalline phase. Therefore, we have added the following statement to the manuscript.
"No obvious selective reflections were observed."
- The measured samples are not described properly. For example, it remains unclear what kind of mechanical grinding was used in samples for data in Fig.3. What is characteristic size of the powder particles after grinding and how it compares with the case of no grinding? How thesamples were prepared, and what was the thickness of the films. Is the film thicknessinfluencing the luminescence spectrum? What kind of substrates were used for spin coating the films. The authors often use terms like "initial substrate", "ground substrate", "annealed and post-grinding substrate" without any description of meaning these terms. I would like to underline that usually the term "substrate" is not the same as the spin coated film (often"substrate" is just a glass plate or a rather thick polymer film used for spin coating a film). Thus it is really difficult to understand what the authors are talking about.
We wrote that we rubbed the thin layer with a pen at discussion part, but we added that method to the experiment section for clarity and we also added that the xerogel powder was rubbed with an agate mortar. We also provide optical micrograph of the luminophore. There were no noteworthy changes in the size or appearance of the powders, and no clear information about the size of the solid powders was obtained, so we believe that changes in the size of the powders are unrelated to changes in the luminescence color. From the previous results, it is known that the change in film thickness does not have a significant effect, and it had no effect in this study.
We used ordinary glass substrates for spin coating, so we added them to the experimental section and changed the expression of "substrates" according to the reviewer's suggestion.
We also tried to measure the layer thickness by using interferometric surface profiler. However, the surface profile of the layer on glass substrate is not uniform and rough chunks are dispersed on the surface. So, we added surface profile image and change several wordings about “films”.
- In case of the directional grinding the mechanical changes of the film remain unclear. What about the film thickness at grinding place - was it changed? Because of the polarized luminescence the question on a birefringence at grinding place arises.
The film thickness change was evaluated by interference microscopy. The xerogels were found to be aggregated and distributed in a speckled pattern. Since the size of the xerogel before and after grinding was undefined, it was not possible to estimate the rough "change in thin layer thickness". Photographs were also added to the manuscript because the luminophore layer showed rather weak birefringence in the ground area. The birefringence is the same as that reported in previous papers.
- The authors describe the photoluminescence as "..a high photoelectron intensity", which isunacceptable. The photoelectrons are just free electrons which appear as a result of interactionwith light (with photons). There are no data showing "photoelectron intensity" in themanuscript, because the photoelectric properties were not measured. There can't be"polarization direction of the photoelectrons". In frame of the work subject, it is correct to speakon orientation of molecular oscillators, electron states, electron transitions and emitted lightpolarization.
The wording has been changed to match the reviewers' suggestions.
- The presented measured data are badly organized into a logical sequence that doesn't allow for understanding or interpretation the described phenomenon in terms of these data.
We reordered the results to clarify the aim of fabrication of polymer composite.

Reviewer 2 Report
The authors have reported a synthesized approach to obtain a liquid crystalline mechano-fluorochromic dye. The manuscript is well written and it is easy to follow. The synthesised molecule is characterized with different techniques and the results show the capabilities of the dye.
Although it is not the core of the manuscript, the electronic structure calculations are poorly reported, and some questions must be clarified. From the theory level: 1) it is not clear if the authors have checked if the optimized structure is a minimum or a transition state; 2) why do they have used B3LYP?; 3) has it been analysed the presence of conformers? 4) what is the phase that the authors have performed the DFT calculations (gas, pcm?). There are some articles that have already studied the molecule, and references will be missing. From my point of view, that sentence will not bring anything new to the manuscript, and my advice is that the authors remove it.
Minor issues:
- Missing ] in line line 25 "[1].
- some other minor typos along the text.
Author Response
Although it is not the core of the manuscript, the electronic structure calculations are poorly reported, and some questions must be clarified. From the theory level: 1) it is not clear if the authors have checked if the optimized structure is a minimum or a transition state; 2) why do they have used B3LYP?; 3) has it been analysed the presence of conformers? 4) what is the phase that the authors have performed the DFT calculations (gas, pcm?). There are some articles that have already studied the molecule, and references will be missing. From my point of view, that sentence will not bring anything new to the manuscript, and my advice is that the authors remove it.
We calculated the minimum state. In general, the electronic states of these dyes are calculated using B3Lyp or CAMB3lyp in the gaseous state, so we followed them.
Phase effects are not considered. As the reviewer pointed out, this study only found the linear shape of the compound (which can be easily expected from the molecular structure) and did not provide any new insights, so this section has been deleted.
Minor issues:
Missing ] in line line 25 "[1].
some other minor typos along the text
We revised the manuscript and highlighted with yellow marker.
Round 2
Reviewer 1 Report
My questions to the first version of the manuscript were addressed. I think, the current version can be published.